# Exploration of recovery of people living with severe mental illness (SMI) in low/middle-income countries (LMICs): a scoping review

Fadia Gamieldien ![ORCID] ,[1,2,3] Roshan Galvaan,[2,3] Bronwyn Myers,[4,5] Zarina Syed,[2] Katherine Sorsdahl[1]

► Prepublication history and additional materials for this paper are available online. To view these files, please visit the journal online (http://dx.doi.org/10.1136/bmjopen-2020-045005).

For numbered affiliations see end of article.

**Correspondence to**
Fadia Gamieldien;
fadia.gamieldien@uct.ac.za

## ABSTRACT

**Objective** To examine the literature on how recovery of people with severe mental illness (SMI) is conceptualised in low/middle-income countries (LMICs), and in particular what factors are thought to facilitate recovery.

**Design** Scoping review.

**Data sources and eligibility** We searched 14 electronic databases, hand searched citations and consulted with experts during the period May–December 2019. Eligible studies were independently screened for inclusion and exclusion by two reviewers. Unresolved discrepancies were referred to a third reviewer.

**Data extraction and synthesis** All bibliographical data and study characteristics were extracted using a data charting form. Selected studies were analysed through a thematic analysis emerging from extracted data.

**Results** The Preferred Reporting Items for Systematic Reviews and Meta-Analyses flow diagram offers a summary of the results: 4201 titles, 1530 abstracts and 109 full-text articles were screened. Ten articles were selected for inclusion: two from Turkey, two from India, and one each from China, Swaziland, Indonesia, Egypt, South Africa and Vietnam. Although most studies used qualitative methods, data collection and sampling methods were heterogeneous. One study reported on service provider perspectives while the rest provided perspectives from a combination of service users and caregivers. Three themes emerged from the data analysis. First, studies frame recovery as a personal journey occurring along a continuum. Second, there was an emphasis on social relationships as a facilitator of recovery. Third, spirituality emerged as both a facilitator and an indicator of recovery. These themes were not mutually exclusive and some overlap exists.

**Conclusion** Although there were commonalities with how high-income countries describe recovery, we also found differences in conceptualisation. These differences in how recovery was understood reflect the importance of framing the personal recovery concept in relation to local needs and contextual issues found in LMICs. This review highlighted the current sparse evidence base and the need to better understand recovery from SMI in LMICs.

## INTRODUCTION

Mental, neurological and substance use (MNS) disorders are significant contributors to the global burden of disease, including

### Strengths and limitations of this study

► A comprehensive search strategy was used which allowed the literature to be mapped as a basis for further exploration of the concept of recovery in low/middle-income countries (LMICs) so as to inform research, policy and practice.

► The review aimed to analyse conceptual and theoretical underpinnings of recovery from severe mental illness (SMI) as a construct in LMICs and as such interventions and intervention outcomes were not focused on.

► Data synthesis was limited to full-text articles available in English only and published between January 1993 and December 2019.

► Grey literature and unpublished studies not available in English were excluded from the scoping review and this practical decision means that potentially relevant papers were excluded.

► Future reviews should search the grey literature extensively as there are non-government organisations active in providing recovery-oriented support and care to people with SMI in LMICs but do not publish in the peer-reviewed literature.

severe mental illness (SMI). The lifetime prevalence of SMI ranges between 1% and 4%.[1] Although this is relatively low in comparison with the prevalence of common mental disorders, people with SMI require complex, long-term interventions.[2] Historically, most resources to respond to MNS were allocated to specialised psychiatric hospitals, with limited provision of mental health services at primary levels of care.[3]

Deinstitutionalisation initiatives in high-income countries (HICs) shifted people out of institutions to community living. The goal of deinstitutionalisation was to promote social inclusion for people with MNS disorders, a goal which remains in progress in low/middle-income countries (LMICs). The drive to decentralise services has led to an increase

in premature discharge rates, shorter hospital stays and repeated relapses.[4–6] This is exacerbated by a scarcity of community-based resources[7] to address the psychosocial needs of service users.[3 4 8]

Personal recovery for people with SMI is conceptually distinct from clinical recovery which places more value on medication adherence and symptom remission.[6 9 10] At a time when having a diagnosis of SMI was highly stigmatised and the clinical prognosis was poor, the recovery model was initiated by the mental health consumer and survivor movement in HICs.[11] The notion that recovery was impossible for people with SMI was challenged by people living with schizophrenia, some of whom went on to lead meaningful lives post-deinstutionalisation.[12 13]

Psychiatric rehabilitation has highlighted the need to manage the functional and disabling consequences of SMI. However, Deegan (a mental health professional and person living with schizophrenia) says that people do not 'get rehabilitated'[11:1] by others, instead they are active participants in their journey to recover new and valued personal meaning and purpose. Personal recovery thus requires a shift beyond treating symptoms to the development of meaning and purpose across the lifespan.[6 14 15]

Recovery means different things to different people, consequently there are multiple interpretations and definitions of recovery. For instance, 17 studies included in a review on the meaning of recovery among people living with schizophrenia,[16] described recovery as both a process and an outcome with multidimensional indicators. Recovery depended on perceptions related to social support, absence of symptoms, minimal to no relapse, regaining regular functioning and resuming responsibilities. This review found that the process of recovery was influenced by acceptance of illness, developing meaning in life, gaining coping mechanisms and regaining functional abilities.[16–18] The complexity embedded in viewing recovery as a process and an outcome with multidimensional indicators has been acknowledged elsewhere.[19–21] Although recovery cannot be defined in a single way, cognisance must be taken that various elements of recovery identified might be common to people's understanding of the experience. Further to this, scholars recommended that clinicians, caregivers and researchers conduct more research to qualitatively explore personal narratives of recovery, develop tools needed to measure recovery and design community-based recovery-oriented services.[16]

Most of the research on personal recovery has emerged from a HIC perspective, with little attention given to describing personal recovery from LMIC perspectives. Of the 17 studies in the aforementioned review,[16] the majority (13) were from HICs (n=13). Similarly, all 97 papers included in a systematic review and narrative synthesis of models of personal recovery[22] originated from just 13 HICs. However, some studies did include ethnic minorities. The value of including perspectives from LMICs and from diverse cultural settings is reflected in the richness and additional themes that emerged from the studies that included ethnic minorities. These additional themes included stigma, spirituality, culture and the collectivist aspects of recovery.[22] Given that recovery is influenced by social, cultural, political, economic factors and respect for human rights, recovery in LMICs is likely to be influenced by sociocultural context.[23] Consequently, a synthesis of the current body of evidence regarding recovery from SMI of people living in LMICs is warranted so as to offer an alternate perspective to the literature predominantly emanating from HICs; this may allow for more diverse perspectives to emerge.[18 24–26] This scoping review thus sets out to explore and report on how recovery is described, and the factors that influence personal recovery of people with SMI in LMICs.

## METHODS

The protocol for this review[27] (online supplemental file 1) provides full details on the methods followed, and the protocol title was registered with the Joanna Briggs Institute. Following Arksey and O'Malley's[25] methodological framework, this scoping review comprised six iterative stages, namely: (1) identification of the review question; (2) identification of relevant studies; (3) selection of studies; (4) charting the data; (5) collating, summarising and reporting of results; and (6) consultation with stakeholders. The Preferred Reporting Items for Systematic Reviews and Meta-Analyses for scoping reviews checklist was used as a reporting standard for documenting the process and results.[25 28]

### Stage 1: identify the research question

We developed a broad research question for our literature search, namely, 'What is known about recovery from SMI in LMICs?'

### Stage 2: identify the relevant studies

A detailed description of the inclusion and exclusion criteria for study selection has been published[27] (online supplemental file 1). Only studies published between 1 January 1993 and December 2019 were included. This covers the roughly 25 years of scholarship since Anthony's seminal work on defining personal recovery from mental illness.[14]

To assist us with accessing relevant publications, we consulted two mental health service providers working in the field of recovery while conducting an electronic search. A comprehensive search strategy[27] was developed by the first author in collaboration with two librarians (MS and DB). The search strategy and filtering methods were tested using preliminary search terms to comply with searches across different databases. The main filtering methods related to the date range (January 1993–December 2019) (online supplemental file 2).

Databases described in the protocol were searched.[27] Grey literature sources were also pursued through Google Scholar searches, contacting study authors and connecting with key personnel involved in recovery-focused

programmes in LMICs. Reference lists of included full-text papers were hand searched for additional sources. Policy documents, conference abstracts, reviews, opinion pieces and commentaries were excluded although they were reviewed to identify published literature.

### Stage 3: literature selection

We followed two independent screening levels for selecting studies for inclusion: (1) title and abstract review; and (2) full-text review. For the first level of review, all citations from the database searches were downloaded into EndNote[29] and duplicates were removed. Thereafter, all citations were imported into Rayyan QCRI software[30] which allowed reviewers (FG and ZS) to screen and select titles and abstracts independently according to the inclusion and exclusion criteria. Inter-rater agreement was assessed by calculating Cohen's kappa. A score of 0.89 was attained, suggesting good inter-rater reliability.[31]

One hundred and nine articles (n=109) were selected for full-text review and assessed to determine if they met criteria for study inclusion. Disagreements on study selection were minimal (n=10) and based on interpretations of the outcome, diagnosis or intervention type described in the article. Disagreements were resolved through consensus seeking with a third rater (KS). At this level of screening, inter-rater agreement was calculated as 0.86. Figure 1 summarises the literature search and selection process.

### Stage 4: charting the data

All 109 articles selected for full review were read. The two reviewers (FG and ZS) independently extracted and summarised data based on Joanna Briggs Institute data extraction template.[24 25 32] Extracted information included: (1) study title, (2) author, (3) year of study, (4) country, (5) study population, (6) participant

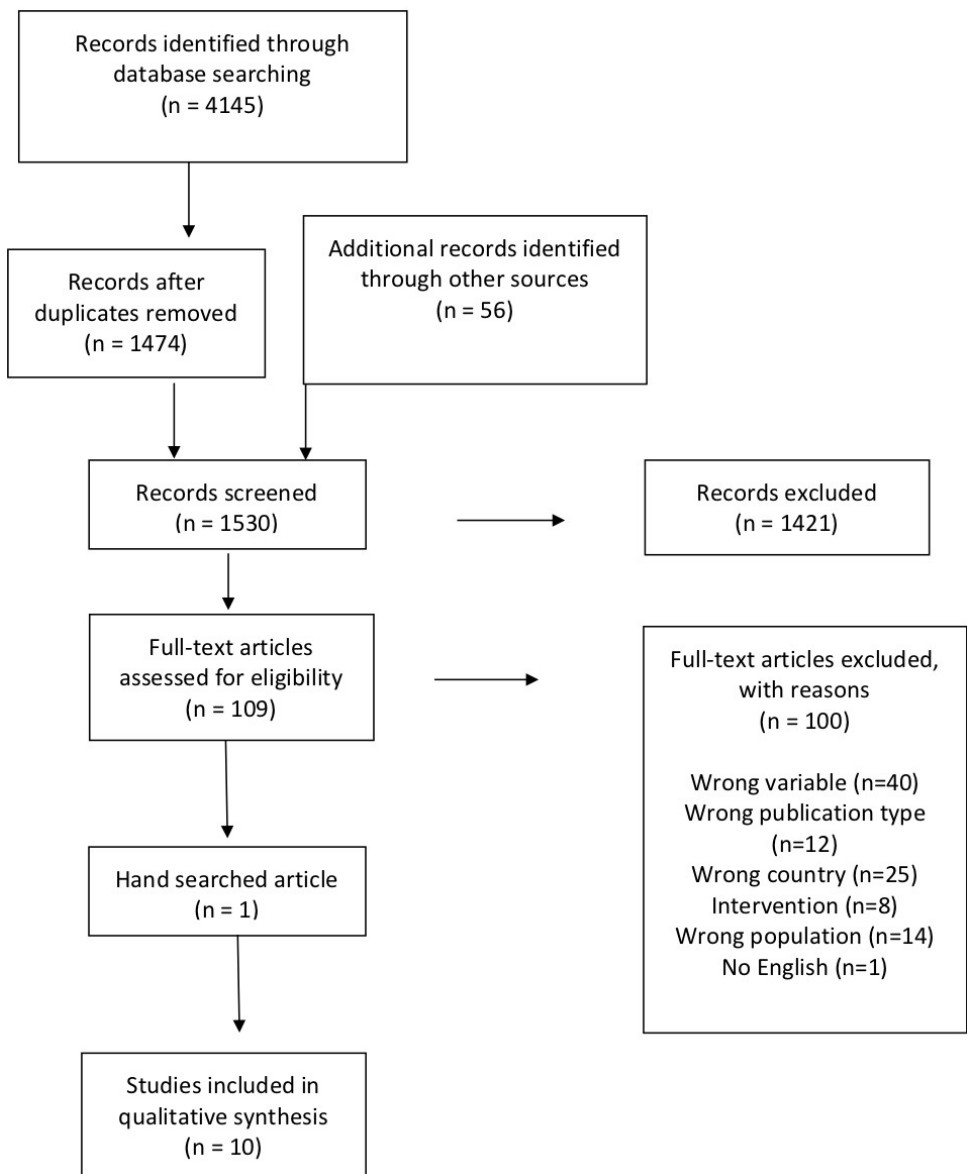

**Figure 1** Flow diagram: Preferred Reporting Items for Systematic Reviews and Meta-Analyses (PRISMA)/PRISMA extension for scoping review.[74]

 

characteristics, (7) number of participants, (8) gender distribution, (9) research question and aim, (10) study methodology, (11) study description, (12) data collection methods, (13) data analysis, (14) outcome measures, (15) results, (16) summary of findings, and (17) definitions and conceptual links of recovery. Online supplemental file 2 provides information on the questions applied when charting the data. To ensure rigour, there was flexibility regarding emerging categories for data extraction. Ongoing consultation with the team occurred throughout the process.

The Mixed Methods Appraisal Tool V.2018[33] and the Critical Appraisal Skills Programme qualitative checklist[34] were used to assess the quality of the included studies. Additionally, we assessed compliance to ethical standards as part of the quality appraisal process. Indicators included (a) obtaining human research ethics before study commencement, (b) documentation of consent to participate and (c) management of vulnerability given that the focus was on people living with an SMI.

### Stage 5: collating, summarising and reporting the results

Collated data were extracted and summarised into a descriptive and narrative synthesis of study characteristics that addressed the review questions. Data were summarised to answer the following questions from LMIC settings: how recovery is conceptualised; what the descriptors of recovery are and what influences recovery. FG completed a qualitative data-driven thematic analysis where findings were coded in qualitative software as recommended by Levac et al[35] using QSR's International NVivo V.11 data analysis software.[36] The coding process was recorded in a memo and reviewed by ZS and KS. All authors contributed to reporting the summarised results.

### Stage 6: consultation

During each stage of the review process, the authors consulted with an advisory committee. This advisory committee consists of the research team and key stakeholders including mental health service providers and service users from the public health and non-profit organisation sectors. This committee was constituted from the onset of the project and some members are part of a preexisting group of public mental health specialists. The advisory committee was consulted to gain their opinion on the relevance of the review. They guided access to grey literature, offered their perspectives on the data extraction and charting process, and reviewed the findings. We included the inputs of an advisory committee in this review to enhance its relevance and support collaborative efforts to facilitate consumer participation and public involvement in mental health research.[32 35]

### Patient and public involvement

There was no direct patient involvement in this scoping review. Mental health service user representatives were involved in designing the scoping review protocol and were part of the advisory committee.

## RESULTS

### Search findings

Of the 1530 articles originally identified, 109 articles were selected for full-text review. Ten were deemed relevant to the research question. Figure 1 outlines the study selection and exclusion process.

The results are presented in two sections: (1) a summary of characteristics and quality of the included studies, and (2) the results of the thematic analysis.

### Description of included studies

Table 1 summarises the 10 studies included in this review. All studies were published between 2014 and 2019 and were conducted in: Turkey (n=2), India (n=2), China (n=1), Swaziland (n=1), Indonesia (n=1), Egypt (n=1), South Africa (n=1) and Vietnam (n=1). According to the World Bank classification system, this covered the range of LMICs. Four studies were conducted in upper middle-income countries, one in a middle-income country and five in lower middle-income countries.

The studies followed different study designs (qualitative: n=8; quantitative n=2). Qualitative studies were guided by the following approaches: grounded theory (n=1), phenomenology (n=3), ethnography (n=2) and qualitative descriptive (n=2). In all the qualitative studies, data were collected through semistructured and unstructured interviews. Five studies focused solely on mental health service users (MHSUs) perspectives,[37–41] one described psychiatrists' perspectives,[42] and three offered a combination of MHSU and family members' perspectives.[43–45] One study reported on multiple stakeholder perspectives by including a combination of MHSUs, family members, community members and healers.[46] Although there was heterogeneity across studies in terms of participants, MHSUs' perspectives were central to conceptualising recovery in most studies.

All participants were adults and the majority of studies focused on a diagnosis of schizophrenia (n=8), while two studies looked at first episode psychosis.[41 44] The age of participants varied between 21 and 70 years, participants had been living with an SMI for between 1 and 40 years and recovery was described as occurring over time.[37–40 43 45 46] The participant's gender profile was described in six studies.[37–40 43 45] In studies where the gender profile was reported, more men were included in the service user groups with the exception of one study which had more women.[39] One study included only female service users.[40] In the two quantitative studies, the majority of caregivers were women.

MHSUs in all of the studies were recruited either directly or indirectly from psychiatric services (n=7) or schizophrenia associations (n=2), with inclusion criteria stipulating a Diagnostic and Statistical Manual of Mental Disorders or International Classification of Diseases, Tenth Revision coding diagnosis. Two studies stipulated that medication adherence was required for participation in the study.[37 38]

**Table 1** Studies included in the review (n=)

| | Authors (year) | Aim(s) | Country; setting according to World Bank classification; recruitment site | Design and methods | Participants |
|---|---|---|---|---|---|
| 1 | Chen *et al* (2018)[45] | To develop an instrument to examine the opinions of consumers and their family members regarding autonomous decision-making in family matters | Chengdu, China<br><br>Upper middle-income country<br><br>Psychiatric hospital | Quantitative<br><br>Descriptive | MHSUs: n=182<br>Women: n=85<br><br>Men n=97<br>Family members: n=182<br>Women: n=103<br>Men: n=79 |
| 2 | de Wet *et al* (2015)[41] | To investigate how MHSUs with first episode psychosis experience their recovery | Cape Town, South Africa<br><br>Upper middle-income country<br><br>Psychiatric hospital research unit | Qualitative<br><br>Descriptive | MHSUs: n=7<br>Gender not reported |
| 3 | Gandhi *et al* (2019)[39] | To explore patients' perspectives about factors affecting recovery from schizophrenia | Bengaluru, India<br><br>Lower middle-income country<br><br>Psychiatric hospital (outpatient department) | Qualitative<br><br>Grounded theory | MHSUs: n=18<br>Men: n=7<br><br>Women: n=11 |
| 4 | Gopal *et al* (2019)[43] | To identify clients and family members' perspectives of what constitutes recovery from schizophrenia in the Indian context and to examine gender differences in recovery indicators | Chennai, India<br><br>Lower middle-income country<br><br>Psychiatric hospital (outpatient department) | Quantitative<br><br>Descriptive | MHSUs n=100<br>Men: n=55<br><br>Women: n=45<br><br>Caregivers: n=80<br>Men: n=43<br>Women: n=37 |
| 5 | Guner (2014)[37] | To understand patients' views and experiences of schizophrenia | Istanbul, Turkey<br><br>Upper middle-income country<br><br>Schizophrenia association | Qualitative<br><br>Descriptive | MHSUs: n=9<br>Men: n=8<br><br>Women: n=1 |
| 6 | Humphries *et al* (2015)[42] | To understand psychiatrists' views on outcomes for people with schizophrenia in a developing country | Danang, Hanoi, Ho Chi Minh City and Hue, Vietnam<br><br>Middle-income country<br><br>Public hospitals | Qualitative<br><br>Descriptive | Psychiatrists: n=15 |
| 7 | Nxumalo Ngubane *et al* (2019)[40] | To explore the experiences and meanings of recovery for Swazi women living with schizophrenia | Swaziland<br><br>Lower middle-income country<br><br>Psychiatric hospital (outpatient department) | Qualitative<br>Phenomenological | MHSUs: n=15<br>Women only |
| 8 | Rashed (2015)[46] | To explore the use of spirit-based frameworks to take control over symptoms in their cultural contexts | Dakhla, Western Desert<br><br>Egypt<br><br>Lower middle-income country | Qualitative<br>Ethnographic | MHSUs: n=2<br><br>Men: n=1<br><br>Women: n=1 |

| | Authors (year) | Aim(s) | Country; setting according to World Bank classification; recruitment site | Design and methods | Participants |
|---|---|---|---|---|---|
| | | | General hospital with psychiatric service | | Family members: n=not specified |
| | | | | | Healers: n=11 |
| | | | | | Community members: n=56 |
| 9 | Soygur et al (2017)[38] | To identify the factors contributing to recovery, as observed from the perspectives of patients with schizophrenia working at a supported employment project | Ankara, Turkey<br><br>Upper middle-income country<br><br>Schizophrenia association | Qualitative Phenomenological | MHSUs: n=24<br>Men: n=15<br><br>Women: n=9 |
| 10 | Subandi (2015)[44] | To explore participants' experience of illness and recovery in a Javanese cultural context | Yogyakarta, Java<br><br>Lower middle-income country<br><br>Psychiatric hospital | Qualitative Ethnographic | MHSUs: n=7<br>Family members: n=not specified |

MHSUs, mental health and substance users.

## Themes from included studies

Table 2 provides a summary of the key findings of each of the 10 included studies. The thematic analysis yielded three themes describing recovery as a personal and complex social process involving healthcare providers, family and community members.[37–43 45 46] These themes, namely (1) recovery is a personal journey occurring along a continuum, (2) aspects of social relationships supportive of recovery, and (3) finding meaning and hope through spirituality and religion are elaborated below.

### Recovery is a personal journey occurring along a continuum

The majority of studies included in this review conceptualised recovery as an ongoing, non-linear process rather than as an event with a finite end.[37–40 43 44 46] The age at onset of illness and duration of illness was highlighted[37–40 43 45 46] as being a signal of when the recovery process began. Onset of illness was usually noticed by family members first and they facilitated health-seeking behaviour on behalf of the MHSU.

Some studies[40 42 44–46] defined the starting point for the recovery process as being from the time that people were back in their homes and communities and not from the start of their admission to hospital. This was confirmed by the methodological decisions taken in all of the included studies to not include inpatients or make reference to inpatient services as facilitating recovery. One study described an outpatient service where the programme included psychosocial skills training, family and group therapy sessions, and group outings.[37] It is notable that recognising symptoms as cues of relapse was not explored since it appeared that the emphasis was on managing symptoms within the home and community settings after the onset of illness.[39 41 43] This approach occurred after a diagnosis had been established for affected MHSUs.[40 42 44–46]

Since the process of recovery was described from the point of discharge, coping strategies related to managing symptoms of their illness and their reintegration within their families and communities emerged as critical facilitators to recovery. MHSUs needed strategies to facilitate this recovery process. Strategies identified in the studies included the development of self-awareness; being able to self-regulate; spirituality including prayer; attending to basic needs; understanding the mental illness; having purpose and hope for the future; gaining a sense of autonomy; engagement in meaningful occupations; contributing to community and adopting a positive attitude to living with a mental illness.[37–41 43 45 46] The studies did not describe the development of these coping strategies. It was also noted that learning to self-manage facilitated the recovery process.[37 40 44] Key aspects that MHSUs had to learn to self-manage included their temperament, learning to live with and derive meaning from having a mental illness and being hopeful for their future.

There was a lack of consensus across the studies regarding the use of medication as an indicator of recovery.[37–40 42–46] Some studies emphasised treatment adherence as the first step towards recovery,[37–40 42 45] while others viewed the choice to stop taking medication as an indicator of recovery.[43 44 46] When describing recovery as it relates to the medical model and clinical recovery, studies appeared to focus on the individual and described contemporary treatment as including a combination of access to medication, health personnel, mental health services at health facilities, and psychosocial interventions.[39 42] According to Rashed,[46] clinical

**Table 2** Summary of findings

| | Title, author, year | Key findings |
|---|---|---|
| 1 | Family Decision Making and Self-Determination Among Consumers With Schizophrenia in China: Cross-Cultural Implications. Chen et al (2018)[46] | ▶ Autonomy in decision-making contributes to the MHSUs functioning during recovery.<br>▶ In Asian contexts, it is accepted practice that family members make decisions on behalf of the MHSUs especially with regard to money management and community and daily living.<br>▶ Decisions related to psychiatric care are often deferred to health professionals.<br>▶ The concept of autonomy for MHSUs includes situations where they endorse and agree with decisions made by family members on their behalf. |
| 2 | Hearing their voices: The lived experience of recovery from first episode psychosis in schizophrenia in South Africa. de Wet et al (2015)[41] | ▶ Caring for others and being cared for was identified as the biggest contributor to recovery of persons with schizophrenia.<br>▶ Spirituality was seen to build resilience more so than adherence to medication.<br>▶ Understanding the mental illness and managing associated stigma influenced disclosure, especially at work. |
| 3 | Perspectives of consumers in India on factors affecting recovery from schizophrenia. Gandhi et al (2019)[39] | ▶ MHSUs in India indicated that recovery was affected by facilitators and barriers related to individual, familial and societal influences.<br>▶ Holding valued roles in a supportive family structure facilitated the recovery of MHSUs.<br>▶ MHSUs reported that spirituality and their engagement in prayer and religious rituals anchored them and contributed towards their recovery.<br>▶ Engaging in activities with others was important for MHSUs as they reported that having meaningful social relationships helped them combat loneliness.<br>▶ Barriers to recovery included side effects of medication, inconsistent treatment approaches and religious beliefs which delayed access to treatment. |
| 4 | What constitutes recovery in schizophrenia? Client and caregiver perspectives from South India. Gopal et al (2019)[43] | ▶ Recovery according to caregivers and MHSUs entailed being symptom free; being able to work and being independent.<br>▶ Recovery was equated with no longer needing medication.<br>▶ Recovery was based on MHSUs' subjective feelings and experiences and not on the opinions of family members and health professionals. |
| 5 | Illness perception in Turkish schizophrenia patients: A qualitative explorative study. Guner (2014)[37] | ▶ Recovery was variably defined and included a lack of symptoms; being able to work; getting married and having opportunities to become part of community networks.<br>▶ MHSUs took ownership for their own personal recovery by engaging in meaningful occupations, hobbies and religious practices including prayers.<br>▶ In Turkey, almost all MHSUs with schizophrenia live with their families.<br>▶ For some, having a supportive family facilitated recovery, but recovery was hindered when the family was overprotective.<br>▶ Recovery was further hindered by difficulties accessing mental health services.<br>▶ MHSUs did not disclose their mental illness for fear of stigma. |
| 6 | Psychiatrists' perceptions of what determines outcomes for people diagnosed with schizophrenia in Vietnam. Humphries et al (2015)[42] | ▶ Access to contemporary treatment was seen as influencing the outcomes for people with schizophrenia. Contemporary treatment is defined as including generic second-generation antipsychotics, staff, facilities and psychosocial interventions.<br>▶ Psychosocial interventions were not described.<br>▶ MHSUs feared stigma and this influenced help-seeking behaviour.<br>▶ Psychiatrists reported that in families who hold traditional beliefs, mental illness was attributed to spiritual causes, families access traditional treatment before seeking medical assistance, especially in rural areas.<br>▶ The extended family system allows family members to share the burden of caring for an MHSU, especially in the absence of formal services. |

Continued

| | Title, author, year | Key findings |
|---|---|---|
| | | ► MHSUs were able to access unskilled work in rural parts of Vietnam more readily than MHSUs in urban settings. |
| 7 | The experiences and meanings of recovery for Swazi women living with 'schizophrenia'. Nxumalo Ngubane *et al* (2019)[40] | ► In Swaziland, MHSUs referred to any mental illness as 'an illness of the brain'. |
| | | ► Schizophrenia is not a known term in Swaziland. |
| | | ► Diagnostic labels such as schizophrenia were used by health professionals but not shared with MHSUs. |
| | | ► Families and significant others provided emotional and financial support to MHSUs. |
| | | ► The presence of therapeutic rapport between MHSUs and health professionals contributed to personal recovery. |
| 8 | From Powerlessness to Control: Psychosis, spirit possession & recovery in the Western desert of Egypt. Rashed (2015)[46] | ► Mental illness was contextualised through spirit-based understanding of the illness and its symptoms. |
| | | ► The relationship MHSUs had with the spirit world was used as a means of describing their recovery journeys. |
| | | ► Clinical recovery was contested as it suggested that mental illness can be cured, whereas personal recovery was described as an ongoing process that did not require medical intervention. |
| | | ► Personal recovery involved MHSUs gaining control over their symptoms. |
| | | ► MHSUs were positioned as being actively involved in their recovery rather than as passive recipients of services. |
| | | ► Spirituality was foregrounded through the religion of Islam and Quranic healing in the lives of people with schizophrenia. |
| 9 | Lessons learned from experiencing Mavi at Café (Blue Horse Café) during six years: A qualitative analysis of factors contributing to recovery from the perspective of schizophrenia patients. Soygur (2016)[44] | ► This study was located in a therapeutic community and supported employment setting. |
| | | ► Long-term relational support between MHSUs and stakeholders involved in health service delivery was seen as being central to recovery. |
| | | ► Recovery was promoted through a number of factors which were embedded in supportive relationships and unrestrictive environments. |
| | | ► MHSUs reported that having work gave them a sense of purpose, responsibility and motivation which added meaning to their lives. |
| 10 | Bangkit: The Processes of Recovery from First Episode Psychosis in Java. Subandi (2015)[44] | ► MHSUs viewed their recovery in the cultural context in which they lived. |
| | | ► Recovery had different aspects to it and involved medical treatment, family care in the home and integration into natural community settings. |
| | | ► MHSUs reported being guided by spirituality which required their active participation in the performance of Islamic religious practices. |

MHSUs, mental health and substance users.

recovery foregrounds biological aspects, implicitly highlighting a curative approach to the illness, often demonstrated through administering medication. However, in exploring psychosis and spirit possession, Rashed[46] found that recovery was not dependent solely on medication or medical intervention. Instead, recovery was conceptualised as an opportunity to expand personal agency in the way MHSUs respond to their illness, with or without medication.[46]

In the study conducted by Subandi,[44] the recovery process involved different phases of participation and integration across social spheres. Recovery as an active process started as MHSUs gained insight into the social isolation that resulted from their mental illness. Social isolation and withdrawal was viewed as the opposite of recovery as it represented inactivity and passivity.[44 46] Across studies, the effort exerted by MHSUs to expand their social interactions from their immediate physical environment to engaging with family members and other stakeholders in health and community spaces was as an indicator of recovery.[37–41 43–46]

From the aforementioned, it can be seen that the people involved in the recovery process included healthcare providers, family and community members. It is also noteworthy that personal recovery required action in a range of interconnected contexts. MHSUs' ability to navigate these different environments indicated what was considered as successful recovery. Indicators of recovery described in the selected articles included: being symptom free, returning to work or supported employment, being independent, getting married, being self-sufficient, making decisions, integrating into social or community networks, and involvement in spiritual and religious activities. Through this, recovery was described as active participation in daily activities. This participation was viewed as an indicator of recovery.[37–46]

### Aspects of social relationships supportive of recovery

Across included studies, the perspectives and contributions of healthcare providers, family and community members were identified as crucial to the MHSUs' recovery process. From these studies, it seems that the ways in which these stakeholders engaged with the recovery process potentially facilitated or hindered the recovery journey.[37–46]

Relational support from health professionals was identified as a facilitator of recovery.[37 38 40 41 45] Access to service providers was influenced by the availability of and access to psychiatric treatment.[42 45] Health professionals supported recovery when they communicated clearly, spent time with service users, offered alternative medication options and provided opportunities for engagement in activities.[37 40] On the contrary, the poor continuity of psychiatric care together with limited and brief consultations with a doctor and a focus on access and adherence to medication was seen to limit the recovery process.[37 39 40 46]

Sociocultural values and variations in explanatory models of mental illness between the MHSUs and health professionals as well as family members influenced views of recovery.[42 45 46] Family members were more likely to seek help from traditional or spiritual healers before seeking contemporary psychiatric treatment.[42 44] The differences between sociocultural values and explanatory models of mental illness included perceived causes of mental illness, traditional beliefs attributing mental illness to spiritual causes, fear of stigma and community attitudes. For example, discrepancies in the language used to refer to mental illness obscured a common understanding of recovery.[40] MHSUs[40] referred to any mental illness as an 'illness of the brain' (p156) while the diagnostic label of schizophrenia was used by health professionals. For one participant, the voice of her grandmother was soothing to her but unacceptable to her family and health service providers.[40] There was an element of dissonance between how the MHSUs, their families and service providers made sense of the mental illness, with a disconnect occurring around how their symptoms were understood by different parties.[46] When the family, community or health

professionals held different understandings and attitudes towards mental illness, recovery was obstructed.

The studies reported that all participants lived with family members or significant others.[37 39–43 46] Although the complexity of large families was acknowledged,[39 42] the extended family supported recovery through sharing the burden of care in the absence of formal health services. The studies highlighted that family and cultural beliefs along with societal values and expectations were used as indicators of recovery.[39 44 45] These indicators were described in different ways and included whether the individual could take care of or provide for their family as they recovered.[37 39–43 46]

Families provided emotional and financial support needed by the MHSUs for daily living during their journey of recovery.[37–44 46] While the family assumed responsibility for decision-making when the MHSUs were acutely ill, they gave up some of this responsibility as the MHSUs recovered.[37 45 46] The families' role as stewards of recovery continued in different ways as they provided flexible access to social networks for MHSUs on an ongoing basis.[37–44 46] Once MHSUs started engaging in spheres such as personal and social functioning, community and daily living and money management, the family's decision-making in these spheres decreased.[40 45]

The importance of social support was recognised as a facilitator of recovery. MHSUs accessed support from different sources which included professionals, family members, therapeutic community networks and supported employment.[37 42 46] Peer support from fellow MHSUs was mentioned once as an important facilitator of recovery.[39] Professional support was symptom related, family support involved assisting with tangible aspects such as financial management or with support for community reintegration. Different networks thus facilitated recovery as they provided relational and emotional support to MHSUs.

MHSUs described having social connections and being able to engage in activities alone and with others as key components of personal recovery.[39 43] The MHSUs' social identity and social position in relation to their family, their community and broader society were seen to influence their functioning and social inclusion.[39 43] Stigma was mentioned as a barrier to recovery but it was not elaborated on in the studies.[37 40–42]

In four studies, MHSUs reported that being able to work facilitated access to social networks.[37 38 40 41] For MHSUs, recovery meant becoming part of a social network where they felt valued and which allowed them to meaningfully contribute to their family and community, whether this was in paid employment or not.[37 38 40 41] One study focused on MHSUs working in a therapeutic community and supported employment setting.[38] The emphasis in this study[38] was on the relational support and social interactions provided to MHSUs via the supported employment project. While employment was important as it gave MHSUs the opportunity to contribute to the financial situation of the family, its role as a facilitator of social

support networks was foregrounded. Recovery as a social process which is influenced positively by collaboration between MHSUs and stakeholders through long-term relational support and encouragement was emphasised.[38]

### Finding meaning and hope through spirituality and religion

Spirituality and religion emerged as resources that the MHSUs and caregivers used for understanding mental illness and the recovery process in 8 of the 10 studies.[37 39–42 44–46] Five studies[37 39 41 42 45] referred to the significance of spirituality and religion. With regard to religion, Islam was referred to in two studies[44 46] and Christianity in one study.[40] Across these studies, spiritual and religious practices were seen to build resilience and were identified as contributing to recovery when they were equally valued by MHSUs, their families and the community.[37 39–42 44–46]

While in recovery, MHSUs underwent internal spiritual struggles when participating in religious practices outside of their homes. The ways in which this was enacted included directly sharing feelings with God through prayer and not through an intermediary, use of religious rituals to relieve stress; surrendering healing to God's will; resisting spirit possession as an act of faith and personal struggle; and by practically supporting places of worship through engaging in cleaning chores as part of church maintenance.[37 39–41 44 46]

MHSUs' relationship with and understanding of the spirit world was seen as an indicator of recovery in one study.[46] Recovery was aided by MHSUs' ability to manage the psychotic symptoms of the mental illness while drawing on the philosophical explanations offered by spirituality. In Egypt, spirit possession featured as part of the experience of psychotic phenomenon and psychosis was not necessarily viewed as a medical problem.[46] According to this study, when MHSUs actively engage in accepted individual and communal religious practices, they are acknowledged as positively navigating personal recovery.

MHSUs reported that spirituality and religious practices provided an anchor for MHSUs and their families as it offered hope for recovery through their own efforts while also having faith and trust in a divine power. Engaging in spiritual practices offered MHSUs structured opportunities to relax, relieve their stress and be hopeful about their future.[39 44] Being hopeful about the future required MHSUs to have faith in themselves and their religion as this allowed them to find peace while living with the challenges of having a mental illness.[37 40 41 46] Furthermore, spirituality was viewed as a resource which contributed positively to the recovery process when MHSUs used it to instil hope for the future and as a mechanism for making sense of the mental illness.

The views held by MHSUs and their family members regarding the influence of spirituality and religious practices on recovery differed from the views held by health professionals.[38–40 42 46] In one study, psychiatrists indicated that religious beliefs and practices could hinder the recovery process particularly where symptoms were attributed to spiritual causes and where traditional treatment was sought before psychiatric treatment.[42] MHSUs viewed professional explanations of mental illness as unhelpful when they discounted the role of spirituality or religion in recovery, especially if they used coercion to support their own ideas of recovery.[40 46] Relying only on medication dispensed by healthcare providers positioned MHSUs as patients and service recipients while adopting a spiritual and cultural interpretation of mental illness, MHSUs were viewed as people who were in control of their illness and who were agents in their own personal recovery process.[44 46] The focus on clinical recovery and the medical interpretation of mental illness minimised the contribution of spirituality to personal recovery.

## DISCUSSION

This review contributes to the growing body of research exploring the meaning of recovery from SMI[15 16 23] by synthesising qualitative and quantitative research from LMICs. No papers from LMICs fulfilled inclusion criteria before 2014, which suggests that the interest in recovery from SMI has been slower in LMICs than in HICs. The 10 articles selected for inclusion were published between 2014 and 2019 suggesting recent and possibly growing interest in understanding perceptions of recovery in these contexts. While there was heterogeneity in terms of design and setting, all studies endorsed the power of individual narrative accounts of recovery. This review highlighted commonalities but also differences between how studies from HICs and LMICs (1) conceptualise the recovery process; (2) outline the individual, interpersonal and social factors that facilitate or impede recovery; (3) define indicators of recovery; and (4) view the role of spirituality and religion in the recovery process.

Findings suggest that recovery for MHSUs is conceptualised as a personal, non-linear and complex process. While this view is echoed in the literature from HICs,[11 16 22 26] the starting point of the process differs between HICs and LMICs. In HICs, MHSUs generally have access to specialised inpatient mental healthcare and their recovery journeys were thought to begin pre-discharge.[47] In the included studies, none of the MHSUs were inpatients at the time of the study; their recovery journeys were described post-discharge. The conceptualisation of recovery as being non-linear in nature incorporates the frequent occurrence of relapses among this population[4] and explains the growing interest in the sociocultural contexts where recovery takes place,[48] particularly in LMICs.

Given the variability in sociocultural contexts, a range of individual, interpersonal and social factors were found to facilitate or impede the process of recovery. First, these publications highlighted the role of explanatory models of mental illness including pathways to care and perceptions regarding the use of medication in the process of recovery. Explanatory models of mental illness[49] provide a framework for understanding SMI and treatment-seeking behaviour in a particular context. Systematic reviews have highlighted

how in many instances traditional healers are often the first point of contact in many LMICs, including African countries.[50 51] Unlike literature from HICs, views were mixed regarding the role of clinical recovery emphasising medication adherence. While some studies highlighted the role of medication in facilitating recovery,[52–54] others emphasised the additional importance of having an awareness of sociocultural and explanatory models of illness as these factors influence MHSUs, caregivers and community members and can either facilitate or impede recovery.[51 55 56]

Second, included studies highlighted how recovery requires long-term relational and emotional support provided by family and friends. The included studies did not draw a distinction between Eastern and Western societies but the findings indicate that sociocultural and religious beliefs favour collectivist perspectives which require the involvement of the family throughout MHSUs' recovery journeys. The role of the family in MHSUs' recovery journeys is emphasised differently in studies from HICs and those from LMICs. In HICs, the deinstitutionalisation process shifted care for MHSUs from hospital to community care. This increased the responsibility placed on family members,[57] but in LMICs access to mental healthcare is a challenge which necessitates that families are involved in the MHSUs' recovery journey from the onset, given the limited availability of community care services.[19]

According to several of the included studies,[37 39 40 42–45] the family members, including significant others, are responsible for the establishment of supportive living arrangements which involved living with family members, another key facilitator in the process of recovery. These living arrangements seem to reflect norms that are embedded in particular cultures globally and were not viewed as a lack of independence or aspiration. In Western society, gaining independence is seen as an indicator of recovery.[11] In contrast, in LMICs, living with supportive family provided MHSUs with opportunities for social cohesion which improved recovery outcomes for MHSUs.[19 58 59] Although supported housing is viewed as a key facilitator of recovery in HICs,[48 60 61] it was not mentioned in any of the included studies described here. Studies in LMICs have examined supported housing for people who are homeless,[62] in the absence of family support. Despite the potentially important role that families can play in the recovery of MHSUs, there is limited literature available on how living with family or extended family facilitates recovery in LMICs.[63]

Similar to HICs, accessing opportunities to build social connectedness and community relationships emerged as a facilitator to recovery. Although research indicates that people with SMI have poorer social connections than the general population,[22] social relationships allow MHSUs to connect with their family, community and work environments. Employment in the skilled or unskilled labour market was found to promote personal recovery of people with SMI.[64–68] Although MHSUs have difficulty accessing employment opportunities, in a number of included studies,[38 40 42–45] the ability to work was described

as a significant indicator of recovery as well as a means of accessing social networks and supportive relationships which facilitated recovery.

Third, spirituality emerged not only as an important facilitator of the recovery process, but a significant indicator of recovery for MHSUs in LMICs. This is in keeping with a review on religion, spirituality and mental health[69] in HICs and LMICs which concluded that engagement in religious practices was a facilitator of recovery. On an individual level, religion was seen as a source of emotional support,[70] but the influence of religion on family and cultural perspectives and practices, even for ethnic minorities in HICs, was not reported on in this review. Included studies in the current review described community religious groups as accessible social networks which provide support to MHSUs and their families.[71] As a facilitator of recovery, spirituality was also seen as a way of finding meaning and hope for MHSUs and their families through the rituals and social connections embedded in practice.[56 72]

The findings of this scoping review need to be considered in light of some limitations. The review aimed to analyse conceptual and theoretical underpinnings of recovery from SMI in LMICs, as such interventions and intervention outcomes were not focused on. Data synthesis was limited to full-text articles available in English and published between January 1993 and December 2019. This practical decision may have led to potentially relevant unpublished papers in other languages being excluded.

## CONCLUSION

This review aimed to examine the literature on how recovery of people with SMI is conceptualised in LMICs, and factors thought to facilitate the recovery process. In this review, we found that understanding of recovery overlapped with those from HICs, but there were important differences. Unlike HICs where independence and autonomy were emphasised, social connectedness and interdependence were emphasised as important indicators of recovery in LMICs.[19] In addition, family support played a much larger role in LMICs than in HICs. Peer-support services from a range of mental health service providers and MHSUs as a strategy for supporting the recovery process are promoted in HICs.[73] Additionally, spirituality emerged as both a facilitator and an indicator of recovery. In HICs, spirituality is rarely considered in the context of recovery, whereas interdependence with the spiritual realm is viewed as an important indicator and facilitator of recovery in LMICs. These differences highlight the importance of assessing whether tools and measures of recovery from SMI, developed in HICs, capture all of the key indicators of recovery that are important for MHSUs living in LMICs. In conclusion, this review highlights the need to expand the recovery concept to be more inclusive of cultural conceptualisations of recovery and illness in LMICs that extend beyond the individual.

**Author affiliations**
[1]Alan J. Flisher Centre for Public Mental Health, Department of Psychiatry and Mental Health, University of Cape Town Faculty of Health Sciences, Rondebosch, Western Cape, South Africa
[2]Division of Occupational Therapy, Department of Health and Rehabilitation Sciences, University of Cape Town Faculty of Health Sciences, Observatory, Western Cape, South Africa
[3]Inclusive Practices Africa Research Group, University of Cape Town, Cape Town, South Africa
[4]Alcohol, Tobacco and Other Drug Research Unit, South African Medical Research Council, Cape Town, Western Cape, South Africa
[5]Division of Addiction Psychiatry, Department of Psychiatry and Mental Health, University of Cape Town Faculty of Health Sciences, Observatory, Western Cape, South Africa

**Acknowledgements** The authors would like to thank the two Faculty of Health Science librarians who assisted with the protocol for this review, namely Mary Shelton and Dilshaad Brey. Their appreciation also goes to Dr Edina Amponsah-Dacosta for her technical insight during the scoping review process.

**Contributors** This scoping review involved the intellectual contributions of all the authors. All authors were involved in developing the review question and conceptualising the approach. FG developed and tested the search terms in consultation with a subject librarian. FG, ZS and KS developed the data extraction guideline, and this was revised based on feedback from the other authors. FG and ZS reviewed all the articles for inclusion and discrepancies were referred to KS. RG, KS and BM provided substantial critique and review of the article prior to submission.

**Funding** This work was supported by the DELTAS Africa Initiative (DEL-15-01). The DELTAS Africa Initiative is an independent funding scheme of the African Academy of Sciences' (AAS) Alliance for Accelerating Excellence in Science in Africa (AESA) and supported by the New Partnership for Africa's Development Planning and Coordinating Agency (NEPAD Agency) with funding from the Wellcome Trust (DEL-15-01) and the UK government. This paper is part of a research project funded by DELTAS, awarded to FG. BM is funded through the South African Medical Research Council.

**Disclaimer** The views expressed in this publication are those of the author(s) and not necessarily those of AAS, NEPAD Agency, WellcomeTrust or the UK government. The funder has no role in the design, writing or submission of the manuscript.

**Competing interests** None declared.

**Patient consent for publication** Not required.

**Provenance and peer review** Not commissioned; externally peer reviewed.

**Data availability statement** No data are available. No additional data available

**ORCID iD**
Fadia Gamieldien http://orcid.org/0000-0003-2820-6484

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
