## [Reviewer comments · BMJ Open]

ARTICLE DETAILS

TITLE (PROVISIONAL)	An exploration of recovery of people living with severe mental illness (SMI) in low- and- middle -income countries (LMIC): A scoping review
AUTHORS	Gamieldien, Fadia; Galvaan, Roshan; Myers, Bronwyn; Syed, Zarina; Sorsdahl, Katherine

VERSION 1 – REVIEW

REVIEWER	Martin Agrest Proyecto Suma, Argentina
REVIEW RETURNED	30-Nov-2020

GENERAL COMMENTS	This is a very good quality paper. Both general conceptualization of recovery and LMIC interest in recovery can benefit from this paper. The Introduction section is good. The Methods section is strong and well delineated. The Results section is good. There is an interesting point made by the authors with regards to when the recovery process is considered to start in LMICs, which is later on adequately incorporated into the Discussion section. The Discussion section is very interesting and makes another good point by comparing how recovery is understood in LMICs and in HICs. Differences between Eastern / Western societies within LMICs may have been interesting to be included in the discussion section (and particularly in connection to individualistic vs. collectivist perspectives, the role of families and the role of religion, if findings support any of these distinctions). It may be interesting to highlight in the Discussion section that no paper fulfilled inclusion criteria prior to 2014, which means there was a significant delay in LMICs to address and become interested in recovery compared to HICs. Minor details require authors' attention: References in the text need to be fixed. The first reference is missing in the text until the Methods section. Different notations are used (e.g., "9 10 11" and "17-19"), references 7, 8 and 9 seem to be out of place (#9 is first in the text and should be #7), etc. Reference #11 has the author duplicated in the reference list and
--

	should be just “Myers NL”. Table #2, on Row #2, the author and date are missing, and a reference was added unlike all other cases in this table. The same criteria should be used in all cases. More important, Table #1 and #2 share the Aims column, and columns for Author & year and for Country are almost identical. Maybe both tables can be merged, or the authors may try to avoid duplications between tables. Peer support (on page 20, line 25) may be confusing as it refers to support provided by others (on equal status) and not to (specialized) “peer support”. The authors would rather be clearer about what type of peer support they are referring to. In the Discussion section (page 22, lines 40-41), the authors highlight four aspects for discussion. However, #3 and #4 overlap. Maybe #4 could just say “view the role of spirituality and religion in the recovery process” since indicators for recovery were analyzed in the previous point (“(3) define indicators of recovery;”).
--	---

REVIEWER	Marta Imamura Instituto de Medicina Fisica e Reabilitacao, Hospital das Clinicas HCFMUSP, Faculdade de Medicina, Universidade de Sao Paulo, Sao Paulo, SP, BR. BRAZIL
REVIEW RETURNED	12-Dec-2020

GENERAL COMMENTS	An exploration of recovery of people living with severe mental illness (SMI) in low- and- middle -income countries (LMIC): a scoping review Authors present a scoping review that aims to examine the literature about recovery from severe mental illness in low- and middle-income countries. Authors followed a sound methodological approach. The scoping review published protocol contains the details of the methodological framework. Authors report the scope review using the PRISMA ScR checklist. Inter-rater agreement of 0.89 was attained for the independent selection of titles and abstracts and of 0.86 for study selection. The Mixed Methods Appraisal Tool and the Critical Appraisal Skills Programme Qualitative checklist were used to assess the quality of the included studies. Authors provided an in-depth and comprehensive discussion on the main findings and highlight the relevant differences identified with the recovery from mental illness in high income countries. Findings described in this scoping review will contribute to the recovery of people living with severe mental illness in low- and middle-income countries.
--

VERSION 1 – AUTHOR RESPONSE

Response to comments	Amendments made	
Reviewer: 1 Dr. Martin Agrest, Proyecto Suma, Community Health Service, Buenos Aries	Thank you for the comprehensive feedback. All comments have been attended to.	
Differences between Eastern / Western societies within LMICs may have been interesting to be included in the discussion section (particularly in connection to individualistic vs collectivist perspectives, the role of families and the role of religion, if findings support any of these distinctions).	Comment added.	Pg.21 The included studies did not draw a distinction between Eastern and Western societies but the findings indicate that socio- cultural and religious beliefs favour collectivist perspectives which require the involvement of the family throughout MHSUs recovery journeys.
It may be interesting to highlight in the Discussion section that no paper fulfilled inclusion criteria before 2014, which means there was a significant delay in LMICs to address and become interested in recovery compared to HICs.	Thank you for this comment. It highlights an important point and is included in the manuscript	Pg. 20 Added: No papers from LMICs fulfilled inclusion criteria before 2014, which suggests that the interest in recovery from SMI has been slower in LMICs than in HICs.
References in the text need to be fixed. The first reference is missing in the text until the Methods section. Different notations are used (e.g., "9 10 11" and "17-19"), references 7, 8 and 9 seem to be out of place (#9 is first in the text and should be #7), etc.	Thank you for this editorial care. The first reference was cited in the Protocol information section. This has been rectified, and the reference list has been amended to reflect references in the manuscript's body only. Discrepancies in notations have been addressed throughout the document. Please direct the authors' attention to any other reference errors or omissions that are related to this comment	
Reference #11 has the Author duplicated in the reference list and should be just "Myers NL".	Noted.	Pg. 24 Reference #10 fixed (Myers NL)
Table #2, on Row #2, the Author and date are missing, and a reference was added unlike all other cases in this table. The same criteria should be used in all	Noted The reference list and in-text citations have been updated for uniformity.	Pg. 12 Table 2 de Wet et al., 2015

cases. More important, Table #1 and #2 share the Aims column, and columns for Author & year and for Country are almost identical. Maybe both tables can be merged, or the authors may try to avoid duplications between tables.	Table 2 has been amended, and the aim and country columns have been removed to avoid duplication across tables. Table 2 reports on key findings for each of the included studies. The Title, Author and Year have been retained for the readers ease of reference.	
Peer support (on page 20, line 25) may be confusing as it refers to support provided by others (on equal status) and not to (specialized) "peer support". The authors would rather be clearer about what type of peer support they are referring to.	Thank you for this comment. Peer support in the included article refers to support from fellow MHSUs.	Pg. 18 Peer support from fellow MHSU Pg. 23 Conclusion In addition, family support played a much larger role in LMICs than in HICs. Peer support services from a range of mental health service providers and MHSUs as a strategy for supporting the recovery process are promoted in HICs.¹
In the Discussion section (page 22, lines 40-41), the authors highlight four aspects for discussion. However, #3 and #4 overlap. Maybe #4 could just say "view the role of spirituality and religion in the recovery process" since indicators for recovery were analyzed in the previous point ("(3) define indicators of recovery;").	Thank you for this suggestion. Point 4 has been amended	Pg. 20 ... and (4) view the role of spirituality and religion in the recovery process.
Reviewer: 2 Dr. Marta Imamura, University of Sao Paulo	Thank you for the positive feedback. It is appreciated.